# Prior Consistent CNN with Multi-Task Learning for Colon Image Classification

Chaoyang Yan[1], Chengfei Cai[1], Jiawei Xie[1], Yao Fu[2], Hui Shuai[1], Xiangshan Fan[2], and Jun Xu[1]✉

[1] Nanjing University of Information Science & Technology, Nanjing 210044, China
[2] Gulou Hospital, Nanjing 210008, China.

**Abstract.** As adenocarcinoma is the most common cancer, the pathology diagnoses for it is of great significance. In the field of digital pathology, although deep learning method has achieved good results, it is theorem agnostic and the accumulated pathology-level knowledge is ignored. Specifically, the degree of gland differentiation is vital for defining the grade of adenocarcinoma. Following this domain knowledge, we encoded gland tissue regions as prior information in a multi-task convolutional neural network (CNN), guiding the network's preference for gland information when inferring. Firstly, we validated the effectiveness of the gland prior information by single task with gland ground truth annotations. Then we constructed a multi-task framework with segmentation and classification branches simultaneously. In this architecture, the segmentation probability map acted as the spatial attention for classification, emphasizing the region of gland and masking the noise of irrelevant parts. Experiments showed that the proposed prior consistent CNN with a multi-task learning method achieved 97.04% accuracy, compared with 93.82% of the single task classification model. Meanwhile, proposed multi-task model outputted gland tissue segmentation results. Most importantly, our model is based on the clinical-pathological diagnostic criteria of adenocarcinoma, which provides more ideas on how to make deep learning methods in the field of digital pathology more interpretable.

**Keywords:** Multi-Task Learning · Gland Prior Attention · Colorectal Cancer · Convolutional Neural Network.

## 1 Introduction

Nowadays, with the development of computer vision (CV), more and more advanced image processing algorithms are applied to the field of medical images[2]. In the field of digital pathology, deep learning plays an increasingly important role because of its excellent performance in image classification, tissue segmentation and cell detection[6]. Usually, deep learning method was simply implemented to process image pixels for final label predictions. Although it achieved or even exceeded human-level performance, it lacks pathology-level interpretation. Also, few studies analyzed inference process of models or considered the domain prior information of pathology to guide model inference once models were designed.

Adenocarcinomas are the most common form of cancer. In pathological diagnosis of several adenocarcinomas including prostate, lung, and colon, the degree of gland differentiation is vital for defining the grade of adenocarcinoma, and it is the important criterion on grading for pathologists[4]. Therefore, CV-based studies of these adenocarcinomas often depend on quantitative description of glands. Xu *et al.* employed a geodesic active contour model for gland segmentation in prostate[10]. Ali *et al.* adopted adaptive active contour scheme with a shape prior for gland-related nuclei segmentation[1]. Also, Zhang *et al.* utilized nuclei shape prior for segmentation by spectral clustering and sparse coding[14]. Important pathological prior knowledge such as gland regions can be segmented, encoded and fused into CNN for constraining, guiding model's inferring preference for tissue prior information and improving pathology-level interpretation.

Multi-task learning is a branch of machine learning, it is an algorithm that can learn multiple tasks at the same time[3]. It improved robustness of features and generalization of models through sharing domain-specific information and joint learning. Yoon *et al.* employed a multi-task CNN for automated extraction of the primary cancer site and its laterality, achieved further performance improvement than single task[11]. Here, we adopted multi-task learning for classification and segmentation, which can improve the performance and generalization of models by sharing the feature representation.

In our work, to utilize strong points of deep learning and improve pathology-level interpretation, we encoded gland regions as prior attention for deep learning network, guiding the model's preference for gland information when inferring. We performed multi-task learning for image binary classification and glands segmentation. Also, we transferred the probability map of automatic gland segmentation to the classification branch for constraining, which achieved benign and malignant discrimination of pathological images based on automated segmented gland prior information.

## 2    Methods

Our proposed prior consistent CNN for classification is based on multi-task learning and gland prior weighted guidance. Fig.1 shows the overall network framework. The entire structure includes a feature learning backbone part and two parallel branches: classification branch and segmentation branch, which are expressed as Branch-C and Branch-S. The backbone part is acted as feature extractor and Branch-C is utilized for making decisions between benign and malignant images, Branch-S is for automatic segmentation of glandular tissue. Branch-S can not only output the segmented results of gland but also can transfer gland's probability map information to Branch-C as prior guidance, which constrains Branch-C on reasoning the category of pathological images. The two branches share features of the backbone part and perform target optimization jointly. We described each module in detail in the following sections.

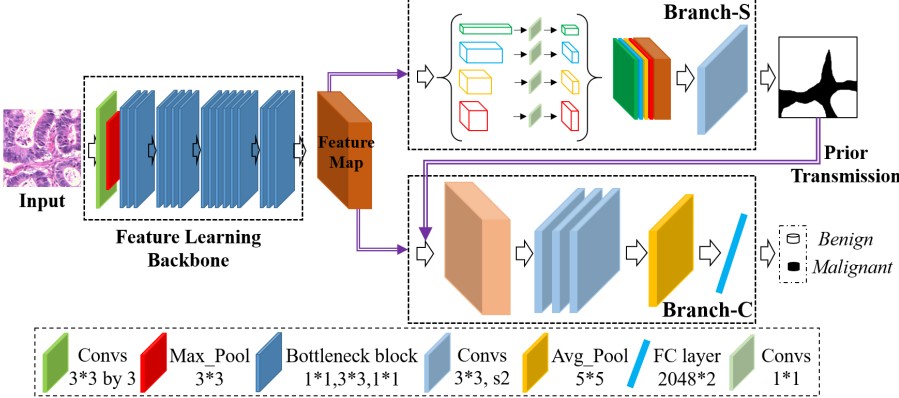

**Fig. 1.** Proposed overall network framework. Our network is composed of a backbone and two parallel branches: classification and segmentation. For an input image, the backbone part is responsible for general feature extraction; Obtained feature maps are then utilized for benign/malignant classification and gland segmentation in Branch-C and Branch-S, separately. Prior transmission is the process of fusing Branch-S output probability map with general feature maps as the input for classification guidance.

## 2.1 Features learning backbone

The Backbone section was adopted for high-level semantic features learning of pathological images through stacking of convolution layers. Here, benefiting from residual network[5], we experienced the backbone part similar to ResNet50[5].

As Fig.1 shown, for an input image $X \in R^{3*h*w}$, 1/4 down-sampling operation is firstly performed by 3 convolution layers with $3*3$ kernels and a max-pooling operation with $3*3$ kernels and stride 2. Then, a 4-part convolution portion contained 3, 4, 6, and 3 residual blocks, respectively, wherein the second convolution portion operated 1/2 down-sampling. In order to obtain greater receptive fields and pay attention to the scale information, we adopted the dilated convolution strategy[13] in the 3rd and 4th portions. Finally, the backbone part outputs feature maps representation $V \in R^{2048*\frac{h}{8}*\frac{w}{8}}$.

## 2.2 Multi tasks with automated segmentation and classification

**Branch-S** was used to implement automatic gland segmentation. In order to improve the ability to obtain context content and global information, the pyramid pooling module (PPM)[15] was utilized. PPM composed of four global pooling operations with outputs of $1*1$, $2*2$, $3*3$, and $6*6$, followed by $1*1$ convolution for channel dimensionality reduction. Then, bilinear interpolation operation was added to obtain the same size features $V'_{bin} = \{V'_1, V'_2, V'_3, V'_6\}$ as $V$. We then concatenated $V$ and $V'_{bin}$ to make model integrative to scale information. At last, there are two convolutions mapping $[V, V'_{bin}]$ features to the final prediction $P$.

The prediction map can be passed to Branch-C as gland prior information and converted to binary gland prediction via bilinear interpolation method.

**Branch-C** was adopted to achieve benign and malignant classification. To utilize gland prior information, weights assignment $W_{prior}$ on the input $V$ was performed, obtaining feature maps $V_w$. Then 3 convolutional operations with $3 * 3$ kernels followed by a global pooling operation and a fully connected layer were added for feature learning and dimension reduction. Finally, the output vectors $v$ were normalized to class probabilities.

**Loss optimization** plays an important role in the multi-task learning network. We minimized the objective function of our model. The overall loss function $L_{MT}$ for each image sample is defined as:

$$L_{MT}(c^*, s^*) = L_{CE}(c, c^*) + \lambda \sum_{j=1}^{w} \sum_{i=1}^{h} L_{CE}(s_{ij}, s_{ij}^*) \quad \lambda \in \{0, 1\}$$

Here, $c^*$ and $s^*$ represent predicted binary probabilities and probability maps of Branch-C and Branch-S respectively. Both branches are optimized by cross-entropy loss, denoted by $L_{CE}$. $c$ is the binary label, and $s$ is the gland ground truth of the corresponding image, where $s_{ij}$ represents the pixel point label of row $i$ and column $j$. The two terms are balanced by hyper-parameter $\lambda$, which we took 0 for the single task classification and 1 for multi tasks with Branch-S.

### 2.3   Gland prior information transmission

Gland prior information transmission is the foundation of reasoning and interpretation, it is also the core of our architecture. It reconstructed the input feature maps of Branch-C by weighted attention encoding and guided the inference.

In this paper, two methods of gland prior information transmission were proposed. One approach is to utilize the Hadamard product $X'$ of the original image $X$ and glandular ground truth annotation $M \in R^{1*h*w}$ as network's input for feature learning, ignoring non-glandular tissue directly. The significance of this method is for verifying effectiveness of gland prior information via the single task classification model. Another method is to constrain the feature maps $V$ via the segmented gland probability map. In that way, not only can we verify the prior effectiveness, but also achieve automated segmentation. Probability map information approximates ground truth content when Branch-S optimization approximates optimal solution; moreover, probability information is in line with the idea of weight attention compared with binary information.

## 3   Experiments Design

### 3.1   Data and preprocessing

Warwick-QU Colorectal Cancer dataset[8, 9] on gland segmentation task was utilized. It includes gland region instance annotations and images diagnostic

information. The entire dataset was divided into the training set with 37 benign and 48 malignant images and test set with 37 benign and 43 malignant images, totally 165 images scanned at 20x magnification.

We cropped 5 patches pairs with a size of $320 * 320$ from each image and its corresponding gland instance annotation. Then gland instance annotations were converted to semantic annotations and each histology image performed color standardization via color transfer[7]. Finally, data enhancement including 3 rotations and 2 mirror operations was carried out for each patch pair. Obtained multi-task learning overall dataset included training set with 984 benign and 1386 malignant and testing set with 1050 benign and 1182 malignant samples.

### 3.2   Stage I, Single task hypothesis validation

The purpose of this experiments stage is to verify the effectiveness of gland prior information. For our multi-task learning network, we first removed Branch-S, and the feature maps output from the backbone was only used as the input of Branch-C, which means that the network structure is a single task classifier. We compared the experimental results of three network models: single task classifier without prior, single task classifier with prior on the image and single task classifier with prior on the feature maps, which we denoted as ST_noP, ST_PoI and ST_PoF. For all experiments of this stage, parameters were 100 training epochs, Adam optimizer, cross-entropy loss, kaiming-uniform weights initialization. The initial learning rate was 0.001 and its decay strategy was 0.5 per 10 epochs. The evaluation indexes were accuracy and AUC values.

### 3.3   Stage II, Multi tasks with automated segmentation

In this stage, Branch-S was reserved. Two tasks including image classification and gland segmentation were achieved by sharing backbone's feature representation. The first experiment in this stage is to carry out the joint optimization of Branch-C and Branch-S without gland prior information transmission. For the second experiment, Branch-C and Branch-S were optimized simultaneously, and the prediction probability map of Branch-S was transmitted to Branch-C. The experimental parameter settings were consistent with the first stage. Also, mIOU index for segmentation was for evaluation.

In addition, due to the kaiming-uniform weights initialization, the initial segmentation results could be introduced into the Branch-C too early, which led to noise. Therefore, we carried out a comparative experiment with pre-trained model for parameter initialization. We denoted these three experiments as MT_noP, MT_P_Uniform and MT_P_Pretrained, respectively. The pre-trained model of MT_P_Pretrained was fine-tuned from trained model of MT_noP.

## 4   Results and Discussion

Table.1 presents an overview of all experiments' performances in terms of accuracy, AUC for classification and mIOU index for segmentation. The top half

shows performances of hypothesis validation experiments of stage I and the bottom half provides stage II evaluation results.

**Table 1.** Overview of all experiments' performances in terms of accuracy, AUC value for classification and mIOU index for segmentation. (ST and MT denote single task and multi tasks, individually. P denotes gland prior attention, I denotes original input image and F denotes backbone output feature maps)

|  | *Experiments Model* | *Classification* | | *Segmentation* |
|---|---|---|---|---|
|  |  | *Accuracy* | *AUC* | *mIOU* |
| *Experiments Stage I* | ST_noP | 93.82% | 0.978 | - |
|  | ST_PoI | 95.70% | 0.9835 | - |
|  | ST_PoF | **96.77%** | **0.9939** | - |
| *Experiments Stage II* | MT_noP | 95.97% | 0.9828 | 0.8086 |
|  | MT_P_Uniform | 95.70% | 0.9901 | 0.7347 |
|  | MT_P_Pretrained | **97.04%** | **0.9971** | **0.8134** |

For stage I, experiments with gland prior both showed better performances than that without gland prior, achieving about 2% and 3% increasing on accuracy, individually. This validated our hypothesis that classification based on gland prior is beneficial. In addition, it is better to transmit gland prior information to feature maps than to the origin image. This performance gain suggested that gland prior on feature maps is equivalent to the attention mechanism[12].

In stage II, MT_noP, MT_P_Uniform and MT_P_Pretrained reached 95.97%, 95.70% and 97.04% on accuracy, respectively. On the one hand, MT_noP model achieved 2.15% increasing than ST_noP on accuracy. This is consistent with that shared backbone part of multi-task learning network can learn more robust features and representation. On the other hand, MT_P_Pretrained model got better performance than MT_noP, more than 1% improvement on accuracy and 0.015 on AUC value. These results further support the hypothesis of importance and effectiveness of gland prior information. Further analysis showed that MT_noP and MT_P_Pretrained models produced almost the same mIOU performances, 0.8086 and 0.8134, respectively. Automatic glandular segmentation can perform well whether gland prior information was transmitted to Branch-C or not. A common view is that Branch-S was not affected although prior information attention was transferred into Branch-C. It is worth noting that no significant improvement on MT_P_Uniform model was found compared with MT_noP. Also, it is apparent from the table that the mIOU of MT_P_Uniform was poor. These indicated that network optimization became difficult due to gland probability map transmission at the beginning of training and the performance was affected because Branch-S is not well optimized and noise was added.

In addition to the evaluation index, we also show some segmentation results of Branch-S in multi-task model. Furthermore, we utilized Class Activation Mapping (CAM) algorithm[16] to generate features and visualize attention maps of Branch-C classifier. Fig.2 shows segmentation results, classification

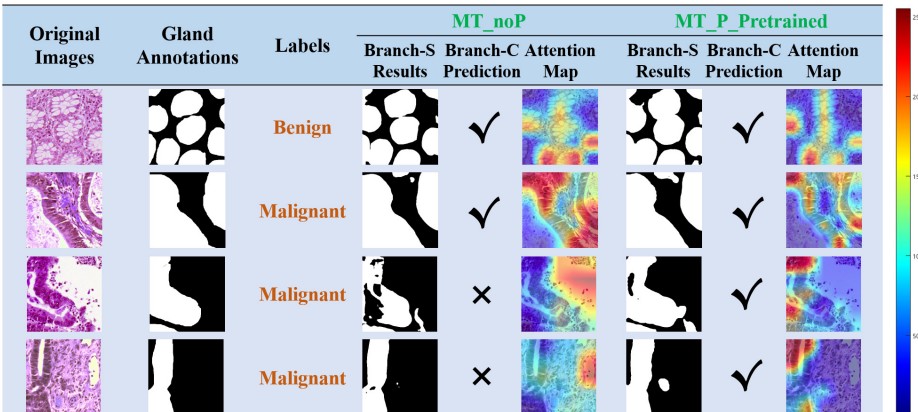

**Fig. 2.** Gland segmentation results, classification predictions and classifier's attention maps for different samples on MT_noP and MT_P_Pretrained models.

predictions and classifier's attention maps of pathological samples on MT_noP and MT_P_Pretrained models. The first two samples were correctly classified by both models, and their gland tissue were well segmented. Also, corresponding attention maps were observed. Branch-C of MT_noP model has focused on glandular tissue although there is no prior constraint. After that, Branch-C of MT_P_Pretrained model paid more attention to glandular tissue content under the constraint of gland prior information. For following two samples, their gland tissue was also well segmented. However, they were misclassified by MT_noP model but correctly predicted by MT_P_Pretrained model. We observed their attention maps. Branch-C of MT_noP model focused on non-tissue regions (lumen and background) thus appeared error recognition. But MT_P_Pretrained model paid attention to gland tissue regions because gland prior information was transferred. The classifier are targeted, that's why it achieved better performance. And this again confirmed the benefits of gland prior weights attention.

## 5   Conclusion

In this work, we proposed prior consistent CNN with multi-task learning for colon image classification, which achieved automated glands segmentation and images classification simultaneously. We encoded segmented gland probability map from Branch-S as prior attention for Branch-C, constraining and guiding Branch-C's preference for gland regions when inferring. Also, we improved the robustness of features by means of feature sharing representation and multi-task joint learning. Our method achieved excellent results on GlaS dataset, compared with single task model. The progressiveness and applicability of our method have been validated. Importantly, our framework was based on the clinical pathological diagnostic criteria, which provides more ideas on how to make deep learning methods in the field of digital pathology more interpretable.

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
