# OpenReview forum: "Prior Consistent CNN with Multi-Task Learning for Colon Image Classification"
_MICCAI.org/2019/Workshop/COMPAY — Submitted to COMPAY 2019_

### Official Review · AnonReviewer1 · 2019-08-13
**Attempt to use clinical prior knowledge in colon image classification**

**Rating:** 5
**Confidence:** 3

**Review:**

This paper proposes a CNN-based approach with multitask learning for simultaneous segmentation and classification of colon images. It aims to incorporate prior knowledge about clinical-pathological diagnostic criteria of adenocarcinomas in grading the images. The proposed approach uses the probability map produced by the glands segmentation branch of the network as prior attention for the classification branch to raise its preference for gland regions when inferring. Both branches make use of a feature map generated by a feature learning backbone.

Methodologically there seems to be very little (if anything) new as the paper draws heavily from other sources for its network design. Perhaps the main contribution is in the use of the segmentation to drive the classification in combination with the particular application in colon image classification.

Experimentally the results are a bit hard to value. While the presented numbers seem promising for the task, it is unclear how they relate to the state of the art in the field. Only different variants of the authors' own network design are compared. But how to they compare to those used in the GlaS challenge, in terms of both design and performance?

Minor:

- Figure 1 could be better explained. The meaning of some symbols such as the different Branch-S blocks is not clear.

- Section 2: No motivation is given why the authors chose the specific network design and parameter settings mentioned. It all comes out of the blue.

- Section 4: "MT_noP, MT_P_Uniform and MT_P_Pretrained reached 95.97%, 95.70% and 97.04% on accuracy, respectively." There is no need to repeat information from Table 1.

---

### Official Review · AnonReviewer4 · 2019-08-13
**Multi-task CNN for classification and segmentation, unclear how this compares with state of the art**

**Rating:** 6
**Confidence:** 5

**Review:**

This paper presents an approach based on CNNs to classify colon patches as containing benign or malignant glands, as well as producing gland segmentation. The model is built based on a multi-task learning approach, where a backbone CNN is used to extract features used to feed two branches, one for classification and one for segmentation tasks. Notably, the output of the segmentation task is used as part of the input to the classification task.

Major comments
* Some parts of the paper are a bit difficult to understand, mostly because of the way content is presented. For example, section 2.3 talks about gland prior informaiton transmission, which apparently is the main contribution os this work. However, it is only briefly explained and not in details. I feel like this should have been explained before in the paper, and more importance given to this section, and more details.
* From the paper it is also not clear what the novelty of the proposed approach is, compared to previous work on multi-task learning and on colon gland segmentation.
* Since data from the public GlaS challenge were used, results from other participants should be reported and compared with the current work.
* In one of the experiments (the ST_PoI setting), the Hadamard product (i.e., pixel-wise multiplication) is used to mask out non-glandular pixels of the input image or of feature maps. What is the reasoning behind this, and how is this applied at test time? Simply the test image is passed as it is? By doing this, the distribution of feature maps is different between training and testing, so authors should explain or at least discuss why they thing this is still expected to work and actually improve performance.
* How is the label for the classification branch picked? Is it the label of the central pixel?
* Authors show an improvement of "more than 1% improvement on accuracy and 0.015 on AUC value". The statistical significance of this difference should be tested.

Minor comments
* Fig.1: using rectangles to show layers is not intuitive, they should represent feature maps, so authors should use arrows for layers and rectangles for feature maps.